# Assessing the reporting of Dengue, Chikungunya and Zika to the National Surveillance System in Colombia from 2014–2017: A Capture-recapture analysis accounting for misclassification of arboviral diagnostics

**Mabel Carabali**[1]*, **Gloria I. Jaramillo-Ramirez**[2], **Vivian A. Rivera**[3], **Neila-Julieth Mina Possu**[4], **Berta N. Restrepo**[5], **Kate Zinszer**[6,7]

**1** Department of Epidemiology, Biostatistics and Occupational Health, McGill University, Montreal, Quebec, Canada, **2** Faculty of Medicine, Universidad Cooperativa de Colombia, Villavicencio, Colombia, **3** School of Public Health, Universidad del Valle, Cali, Colombia, **4** Clinton Health Access Initiative, Tegucigalpa, Honduras, **5** Instituto Colombiano de Medicina Tropical- Universidad CES, Medellín, Colombia, **6** School of Public Health, University of Montreal, Montreal, Quebec, Canada, **7** Centre de Recherche en Santé Publique, Montreal, Quebec, Canada

* mabel.carabali@mail.mcgill.ca

## Abstract

### Background

Chikungunya, dengue, and Zika are three different arboviruses which have similar symptoms and are a major public health issue in Colombia. Despite the mandatory reporting of these arboviruses to the National Surveillance System in Colombia (SIVIGILA), it has been reported that the system captures less than 10% of diagnosed cases in some cities.

### Methodology/Principal findings

To assess the scope and degree of arboviruses reporting in Colombia between 2014–2017, we conducted an observational study of surveillance data using the capture-recapture approach in three Colombian cities. Using healthcare facility registries (capture data) and surveillance-notified cases (recapture data), we estimated the degree of reporting by clinical diagnosis. We fit robust Poisson regressions to identify predictors of reporting and estimated the predicted probability of reporting by disease and year. To account for the potential misclassification of the clinical diagnosis, we used the simulation extrapolation for misclassification (MC-SIMEX) method. A total of 266,549 registries were examined. Overall arboviruses' reporting ranged from 5.3% to 14.7% and varied in magnitude according to age and year of diagnosis. Dengue was the most notified disease (21–70%) followed by Zika (6–45%). The highest reporting rate was seen in 2016, an epidemic year. The MC-SIMEX corrected rates indicated underestimation of the reporting due to the potential misclassification bias.

**Data Availability Statement:** Data cannot be shared publicly because we included capture data from medical registries of subjects who sought care at the healthcare institutions and contained their national unique identification number to be linked to the surveillance data. Surveillance data are available from the local health offices of each city upon formal request to the local authorities in each site: Secretary of Health for Cali (https://www.cali.gov.co/salud/), secretary of Health for Medellin (http://www.medellin.gov.co/salud) and Secretary of health for Villavicencio (http://www.villavicencio.gov.co/NuestraAlcaldia/Dependencias/Paginas/Salud.aspx). All other relevant data are within the manuscript and its Supporting Information files. Any additional request please contact Mabel Carabali (mabel.carabali@mail.mcgill.ca) or Kate Zinszer (kate.zinszer@umontreal.ca).

**Funding:** The funding for this work was provided by the Public Health Research Institute from the University of Montreal (seed grant competition). MB holds a CIHR-Banting-Best Doctoral Award fellowship. The funders had no role in study design, data collection and analysis, decision to publish, or preparation of the manuscript.

**Competing interests:** The authors have declared that no competing interests exist.

## Conclusions

These findings reflect challenges on arboviruses' reporting, and therefore, potential challenges on the estimation of arboviral burden in Colombia and other endemic settings with similar surveillance systems.

## Author summary

Colombia is one of the countries most affected by dengue, chikungunya, and Zika in the Americas and are diseases of mandatory reporting to the national surveillance system. Surveillance systems are vital components of disease control programs to understand disease burden, trends, and to detect outbreaks. However, underreporting can bias estimates and greatly reduce surveillance data utility. We used a capture-recapture analysis to identify cases that were diagnosed at the healthcare facilities (capture) and those that were reported to SIVIGILA (recapture) in order to assess the degree of overlapping between diagnosis and reporting. Our study was conducted in three different endemicity settings and showed that the overall reporting of arboviruses was variable in magnitude and dependent on several factors including age of the case, year of diagnosis, and type of health care provider. We also identify the need to correct for misclassification when estimating arboviral burden based on clinical diagnosis in endemic settings using data from passive surveillance systems.

## Introduction

Dengue, chikungunya, and Zika are three different arboviruses primarily transmitted to humans by *Aedes* mosquitoes [1–3]. Dengue is the most common arboviral infection, affecting people in more than 120 countries, with over 90 million cases reported every year [2, 4]. Chikungunya was first introduced in the Americas in 2013 with the epidemic peaking in 2015. Following chikungunya introduction, Zika was introduced into the Americas at the beginning of 2015 and declared a public health emergency in 2016 [1, 2, 5–7]. Together, these arboviruses result in substantial economic burdens estimated at US$8.9 billion for dengue each year [8, 9], US$73.6 million for the cost of one large scale chikungunya outbreak in Colombia [10], and US$2.3 billion for the Zika pandemic [6].

The clinical presentation of dengue, chikungunya, and Zika is similar [1]. Distinguishing features include headache and abdominal pain for dengue infection whereas for chikungunya, the symptoms include incapacitating severe joint pain and for Zika, the presence of conjunctivitis and rash [1, 11–14]. A significant portion of infected individuals with chikungunya, dengue, and Zika remain asymptomatic, with reports of up to 30% of asymptomatic infections for chikungunya, 50% for dengue, and 80% for Zika [1, 2, 11, 15]. While most individuals infected with arboviral diseases will spontaneously recover following the febrile stage, there can be severe infections and chronic sequela [1, 5, 11, 16–18]. Lifelong immunity can be developed for chikungunya and for each one of the four dengue serotypes [14, 19–21]. However, secondary or subsequent infections from different dengue serotypes increases the risk of severe dengue, which occurs in approximately 1% of all dengue cases, with mortality rates up to 20% [3, 16, 20]. Other severe outcomes include the congenital Zika syndrome, which is a set of congenital malformations that has been causally related to Zika virus exposure in fetuses [1, 22, 23], and acute flaccid paralysis or Guillain-Barré syndrome that could occur with Zika among

adults [18, 23, 24]. Sequalae can include post-dengue syndrome, lasting up to two years following initial infection [17, 25] and debilitating joint pain or arthritis that can persist for months or years with chikungunya infection [1, 19, 26].

Historically, Colombia is one of the countries most affected by dengue, chikungunya, and Zika in the Americas region with dengue cases being diagnosed and reported to the national surveillance system since 1950 [4, 27–30]. Although confirmation of these arboviruses should be based upon a positive result from antigen, antibody, or virus detection and/or isolation, the majority of diagnosis in endemic and resource constraint settings is mostly based on clinical symptoms [1, 12, 13, 31]. In Colombia, confirmation of probable cases is largely based on clinical diagnosis plus at least one serological positive test or an epidemiological link to a confirmed case 14 days prior to symptoms onset [32, 33]. In 2016, in Colombia, approximately 41% of all cases of dengue, 1% of chikungunya, and 10% of Zika were laboratory confirmed by Enzyme Linked Immunosorbent Assays (ELISA) or polymerase chain reaction (PCR) with the remainder considered confirmed based on clinical criteria [33–35]. However, the sensitivity (80–95%) and specificity (11–89%) of clinical confirmation for chikungunya, dengue, and Zika is highly variable with higher specificity and positive predictive values in areas with high prevalence [1, 11–13, 36, 37].

Epidemiological surveillance programs are essential components of disease control programs to understand disease burden, trends, and to detect outbreaks [4, 7, 27, 37, 38]. Crucially, surveillance data is often used to guide the planning, implementation, and evaluation of health programs and interventions [6, 29, 36, 39]. However, a major challenge with the use of surveillance data is underreporting of the disease or activity under surveillance, which can bias estimates and greatly reduce the utility of surveillance data [1, 3, 4, 36, 37, 40]. Despite the mandatory reporting of dengue, chikungunya, and Zika to the National Surveillance System (SIVIGILA) in Colombia [32, 33], significant underreporting was estimated for different settings in Colombia, with the national surveillance program capturing less than 10% of diagnosed cases [37, 38]. We consider that individual characteristics and aspects related to health care access shape the reporting of this conditions in the Colombian context. To contribute to a better understanding of the reporting of dengue, chikungunya and Zika in Colombia, we conducted a capture-recapture study of the arboviral diseases in three cities of Colombia from 2014 to 2017. In addition, given the possibility of misclassification of the diagnosis of arboviruses in endemic settings, we conducted a sensitivity analysis to explore the extent of misclassification in the reporting of arboviruses in the study settings.

## Methods

### Ethics statement

This study was approved by University of Montreal's IRB board (Project 18-073-CERES-D) and each ethics committee board from the collaborating institutions.

We conducted an observational study of surveillance data, using the capture-recapture approach [40], to examine the scope and degree of reporting of dengue, chikungunya, and Zika cases in the national surveillance system of Colombia from 2014 to 2017.

### Study sites

Our study was conducted in three different cities with varying levels of endemicity. In 2017, Cali reported 14.8%, Medellín 8.4%, and Villavicencio 2.0% of all dengue case notifications in the country [35]. Medellin is the second largest city in Colombia with more than 2.6 million inhabitants, located in Antioquia province in the Centre-west region[41]. Annual dengue incidence in Medellin ranged from 161 to 745 cases per 100,000 inhabitants during the last 10

years[27, 30, 42]. Cali is the third largest city in Colombia with a population of 2.4 million inhabitants, located in Valle del Cauca's province in the Southwestern region of the country [41]. Annual dengue incidence in Cali ranged from 243 to 692 cases per 100,000 inhabitants during the last five years and 25% of all chikungunya cases reported in the country were from Cali[27, 29, 30]. Villavicencio is the capital of Meta province, with 531,275 inhabitants and located in the Centre-east region[41]. Annual dengue incidence in Villavicencio ranged from 99 to 885 cases per 100,000 inhabitants during the last five years and 3.2% of all chikungunya cases reported in the country were from Villavicencio[29, 30].

## Design and data sources

The capture-recapture method allows the evaluation of overlap between different data sources for disease monitoring [40, 43]. The cases are 'captured' by one data source and are 'recaptured' if they appear in a second data source [40, 43]. The national surveillance program in Colombia receives weekly reports from all health facilities that provide services to probable and confirmed cases of dengue, chikungunya, and Zika [32, 33]. According to the national surveillance protocols, dengue, chikungunya, and Zika should be notified to the national system and cases of severe dengue, dengue-related deaths, and chikungunya-related deaths must be laboratory confirmed [32, 33]. For Zika, laboratory confirmation is mandatory for pregnant women, children <1 year, people with co-morbidities and or neurological symptoms, deaths, stillbirths or children with congenital malformations in endemic areas, if they are suspected cases [33].

We obtained four electronic datasets (two from Cali and one each from Medellin and Villavicencio), with registries of cases seeking care at the different collaborating healthcare facilities and surveillance data (SIVIGILA) from the Secretary of Health of each municipality. We obtained either complete datasets in an Excel format from the collaborating institutions or obtained data extractions based upon a case report form (CRF). The CRF included information on sex, age, ethnicity, reason of consultation, International Classification of Diseases Code 10 (ICD-10) entry and discharges' diagnosis, date of onset of symptoms, treatment, inpatient/outpatient status, and final classification (S1 and S2 Appendices).

Our capture sample was composed of all registries from patients who sought care at the collaborating healthcare facilities between January 1, 2014 to December 31, 2017 and with a healthcare facility's diagnosis, according to the ICD-10 codes of undifferentiated fever, dengue, chikungunya, Zika, or non-specified vector-borne disease. In Colombia, individuals receive healthcare according to a health system coverage which is mainly divided into a subsidized system (for individuals without income or income below the minimum wage for which the government subsidized the healthcare attention) and the contributory system (for employees or individuals who could pay for the healthcare insurance)[29, 32, 44]. There are different healthcare providers or insurer in the country, each with healthcare facilities for different levels of attention within their network, providing care to users of either insurance system[44]. Given the possibility that individuals attend different healthcare facilities and in order to include a comprehensive capture sample, we used centralized data (main consultations and referrals) from the main healthcare providers in each city when available or possible. In Cali, we obtained data from a public/government owned network that provides care mainly to subsidized users in four healthcare facilities and the second dataset included data from all the healthcare facilities providing care to users from one of the main contributory insurance companies in the city. In Medellin, we used data from the main insurer in the city which provides healthcare attention to subsidized and contributory users in 10 main healthcare facilities, we also included data from a facility that mainly provide healthcare to users from the contributory

system. In Villavicencio, we included the data from the main healthcare facility which is as well a university hospital and provide care to users from both subsidized and contributory insurance scheme.

Our recapture sample included all individual cases notified to SIVIGILA as dengue (code 210 or 220), chikungunya (code 217), or Zika (code 895) during the study period in each city and the data was provided by the local health office (S1 and S2 Appendices).

## Data analysis

Descriptive statistics were estimated and presented as median and interquartile ranges (IQR) for continuous variables and proportions for binary or categorical covariates. Using the national unique identification number from healthcare facility registries (capture data), we linked the study records to SIVIGILA's notified cases (recapture data), to assess the degree of reporting by clinical diagnosis with respect to the healthcare facility's registries. Clinical diagnosis was a categorical variable that included the healthcare facility's final diagnosis according to the ICD-10 codes, with the following four levels: "undifferentiated fever", "dengue", "chikungunya", and "Zika/ non-specified vector-borne disease". The latter level was constructed including all cases identified as "Non-specified vector-borne disease" and all Zika diagnosis from the institution, given that at the moment of its introduction, Zika did not have assigned an ICD-10 code [33, 45]. SIVIGILA does not allow the report of "unspecified arboviruses" or "non-specified vector-borne diseases", therefore the institutions used the dengue, chikungunya, or Zika respective code from SIVIGILA to report the cases [32, 33, 45]. Our outcome variable was "reporting" which was a binary variable consisted of cases captured by both systems. Specifically, clinically or laboratory confirmed cases were classified as either: cases identified only in the healthcare facility's registries that were not in SIVIGILA (reporting = 0) or cases that were identified on the healthcare facility registries and also included in SIVIGILA (reporting = 1).

To estimate the level of reporting we followed the approach described by Vong S, et al.[40], where we estimate $N$ as the total number of arboviruses cases as follows:

$$N = \left[ \frac{(N_A + 1)(N_B + 1)}{X_{AB}} \right] - 1$$

Where $N_A$ is the number of cases by the capture (from collaborating institutions), $N_B$ is the number of cases by the recapture (cases notified in SIVIGILA), and $X_{AB}$, is the number of cases reported by both, the capture and recapture. The multiplication factor (MF) is estimated by: $MF = N/N_B$, and is the number by which the surveillance- reported cases needs to be multiplied in order to estimate a relatively valid-true total number of cases, assuming representativeness of the reported cases to the surveillance system[37, 38, 40, 46–49]. In the literature MFs ranges from 1 to 28 for dengue cases according to the year of notification and the type (sever vs non-severe) and could be higher for other emergent diseases [39, 46–51].

To obtain adjusted rates of reporting, we fit site-specific Poisson regressions with robust variance estimation to determine the predictors of the reporting and to estimate the adjusted probability of reporting by clinical diagnosis and year in each study site. We choose robust Poisson models over logistic regression to avoid overestimation of the rates given that the outcome is not rare, the non-collapsibility associated to the estimation of odds ratios, and lack of convergence of the binomial regressions with a log-link [52–54]. The adjusted model included individual level covariates including sex, age, type of diagnostic test, and health care provider (subsidized or contributory) information, when available. Given the possibility of misclassification of clinical diagnosis of arboviruses (e.g., diagnosing chikungunya or Zika cases as

dengue at early stages of its introduction) [1, 11, 13, 15, 29, 36], we conducted a sensitivity analysis using simulation extrapolation for misclassification MC-SIMEX [55, 56]. In brief, the MC-SIMEX method is a tool developed to correct or assess the extent of the misclassification bias. The assessment and correction of the bias includes the extrapolation of the observed regression results from the adjusted model via simulation and correcting the reporting estimates using a correlation matrix with parameters of sensitivity and specificity for each possible diagnostic. In the simulation step, a degree or range of measurement error is applied to the naïve estimates (observed data), according to the provided information of specific/sensitivity of the observations. In the extrapolation step, the results are extrapolated back to an expected point of absence of measurement error [55–57]. To test the robustness of the stimulation estimates, the MC-SIMEX procedure in this study included the same sets of three correlation matrixes for each city: 1) a low sensitivity/specificity matrix, were we set the correlation matrix between main diagnosis to 70%; 2) a high sensitivity/specificity matrix, were we set the correlation matrix between main diagnosis to 90%; and 3) a correlation matrix using the observed data, this is, the actual correlation observed between the notified diagnosis and the institutional diagnosis, having the clinical diagnosis as the reference (S3 and S4 Appendices). Parameter values of sensitivity/specificity were informed by published reports in the literature and estimates from confirmed cases of analyzed data [12, 37]. All analyses were conducted using R Studio (R version 3.6.1, R Core Team; Vienna, Austria;2019).

## Results

A total of 266,549 cases and their related information were obtained from both the capture (healthcare facility) and recapture (SIVIGILA) data across the three study sites between 2014 to 2017. The distribution of dengue, chikungunya, and Zika varied across the study sites and year of reporting. Among the three sites, Cali had the largest number of arboviral cases notified in SIVIGILA (n = 75,379). During the study period, the largest number of SIVIGILA's notified cases was observed in 2016 in Cali and Medellin (43% and 65% of the total cases notified, respectively), while Villavicencio had 80% of the notified cases evenly distributed between 2014 and 2015. There were no differences in sex or in the median age distribution between capture and recapture data (Table 1). However, cases from the capture data (healthcare facility) were slightly younger than the cases among the recaptured data (SIVIGILA) in Medellin and Villavicencio.

The overall crude proportion of reporting was 11.5% (n = 11,223/97,193) in Cali, 14.7% (n = 4,760/14,734) in Medellin, and 5.3% (n = 961/18,121) in Villavicencio. The degree of agreement between healthcare facility's diagnosis and SIVIGILA's notified diagnosis changed by city and arboviral condition. The highest diagnosis' agreement was for dengue, ranging from 49.1% in Medellin, 53.9% in Villavicencio, and 64.0% in Cali. Across all study sites, the surveillance diagnosis of Zika were mostly reported in the institutions as non-specified vector-borne diseases or undifferentiated fever (Fig 1 and Table 2).

The overall -unadjusted- multiplication factors (MF) obtained from the capture-recapture analysis following Vong's approach were 8.7, 6.8 and 18.9 for Cali, Medellin and Villavicencio, respectively. Across arboviruses, the lowest unadjusted MFs were for dengue: MF = 2.5 in Cali; MF = 5.9 in Medellin, and MF = 2.03 in Villavicencio; while the highest were for chikungunya in Cali (MF = 38.2) and Medellin (MF = 8.6) and for Zika in Villavicencio (MF = 110.6) (S3 Appendix).

The adjusted analysis showed that individuals over 18 years of age were more likely to be "captured" by both systems (reported), compared to individuals under 17 years of age, in Cali Rate Ratio (RR) = 3.43 (95%CI = 3.24, 3.64) and Medellin RR = 1.51 (95%CI = 1.43, 1.60).

**Table 1. Descriptive characteristics of arboviral cases reported by data source in Cali, Medellin, and Villavicencio (Colombia), 2014–2017.**

| Site | Cali | | Medellin | | Villavicencio | |
|---|---|---|---|---|---|---|
| Data source | SIVIGILA (Recapture) | Capture | SIVIGILA (Recapture) | Capture | SIVIGILA (Recapture) | Capture |
| N | 75,379 | 97,193 | 28,689 | 32,437 | 14,734 | 18,121 |
| Age (years), median (IQR) | 28 (15, 43) | 31 (6, 48) | 28 (17, 44) | 17 (4, 37) | 27 (15, 39) | 17 (3, 40) * |
| **Age group** | **n (%)** | **n (%)** | **n (%)** | **n (%)** | **n (%)** | **n (%)** |
| ≤17 years | 21,611 (28.7%) | 36,351 (37.4%) | 7,687 (26.8%) | 16,515 (50.9%) | 4,296 (29.2%) | 9,137 (50.4%) |
| ≥18 years | 53,768 (71.3%) | 60,842 (62.6%) | 21,002 (73.2%) | 15,922 (49.1%) | 10,438 (70.8%) | 8,984 (49.6%) |
| **Sex†** | | | | | | |
| Male | 35,172 (46.7%) | 47,381 (48.7%) | 13,641 (47.5%) | 14,263 (44.0%) | 5,994 (40.7%) | - |
| Female | 40,207 (53.3%) | 49,812 (51.3%) | 15,048 (52.5%) | 18,174 (56.0%) | 8,740 (59.3%) | - |
| **Diagnosis** | **n (%)** | **n (%)** | **n (%)** | **n (%)** | **n (%)** | **n (%)** |
| Undifferentiated fever | - | 52,748 (54.3%) | - | 31,070 (89.1%) | - | 14,000 (77.3%) |
| Dengue | 53,349 (70.8%) | 18,940 (19.5%) | 27,285 (95.1%) | 3,658 (10.5%) | 5,260 (35.7%) | 1,639 (9.0%) |
| Chikungunya | 4,437 (5.9%) | 16,944 (17.4%) | 774 (2.7%) | 64 (0.2%) | 4,900 (33.3%) | 599 (3.3%) |
| Zika | 17,593 (23.3%) | - | 630 (2.2%) | - | 4,574 (31.0%) | - |
| Non-specified vector-borne diseases | | 8,557 (8.8%) | | 74 (0.2%) | | 1,879 (10.4%) |
| **Year** | **n (%)** | **n (%)** | **n (%)** | **n (%)** | **n (%)** | **n (%)** |
| 2014 | 1,220 (12.2%) | 16,908 (20.2%) | 3,375 (11.8%) | 8,351 (25.7%) | 2,330 (15.8%) | 5,580 (30.8%) |
| 2015 | 3,845 (38.5%) | 38,982 (46.5%) | 4,485 (15.6%) | 7,530 (23.2%) | 6,023 (40.9%) | 5,440 (30.0%) |
| 2016 | 4,333 (43.3%) | 22,085 (26.4%) | 18,654 (65.0%) | 11,346 (35.0%) | 5,898 (40.0%) | 3,837 (21.2%) |
| 2017 | 599 (6.0%) | 5,769 (6.9%) | 2,175 (7.6%) | 5,210 (16.1%) | 483 (3.3%) | 3,264 (18.0%) |

* There was no individual information of age from the health facility data from Villavicencio.

† Health facility data from Villavicencio had incomplete information for the variable sex.

Compared to the year 2014, the highest rate of reporting was observed during 2016 in Cali (2.28; 95%CI = 2.22, 2.56) and Medellin (RR = 1.9; 95%CI = 1.77, 2.04). Clinical diagnosis of dengue and Zika/non-specified vector-borne diseases were associated with reporting in the three cities. Dengue was the clinical diagnosis with the highest rate of reporting in Cali (RR = 3.6; 95%CI = 3.43, 3.78), Medellin (RR = 5.45; 95%CI = 5.17, 5.75) and Villavicencio (RR = 8.99; 95%CI = 7.62, 10.61), compared to the clinical diagnosis of undifferentiated fever (Table 3).

The estimated adjusted probability of reporting any given clinical diagnosis, obtained from the regression model, changed by year in each study site (Fig 2). The highest average predicted probability of reporting was observed for dengue, in all study sites: Cali 21.9% (95% CI = 21.3%, 22.4%), Medellin 47.6% (95%CI = 46.1%, 49.2%), and Villavicencio 21.9% (95%

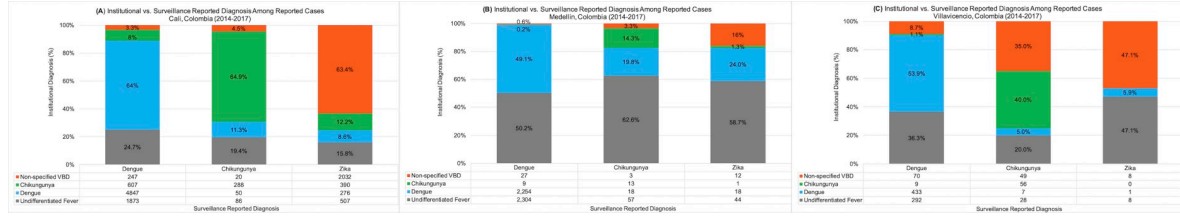

**Fig 1. Agreement between health care facility's diagnosis and SIVIGILA's notified diagnosis of arboviral diseases in Colombia (2014–2017).** Fig 1A: Healthcare facility's diagnosis and SIVIGILA's reporting in Cali. Fig 1B: Healthcare facility's diagnosis and SIVIGILA's reporting diagnosis in Medellin. Fig 1C: Healthcare facility's diagnosis and SIVIGILA's reporting diagnosis in Villavicencio. Non-Specified Vector-Borne Diseases (Non-specified VBDs).

**Table 2.  Agreement between health care facility's diagnosis and SIVIGILA's notified diagnosis of arboviral diseases in Cali, Medellin, and Villavicencio (Colombia) 2014–2017.**

| Capture data | | Cali (n = 97,193) | | | Medellin (n = 32,437) | | | Villavicencio (n = 18,121) | | |
|---|---|---|---|---|---|---|---|---|---|---|
| Reported, n (%) | | 11,223 (11.5%) | | | 4,760 (14.7%) | | | 961 (5.3%) | | |
| **SIVIGILA's reported diagnosis*** | | | | | | | | | | |
| Healthcare facility diagnosis† | | Dengue | Chikungunya | Zika | Dengue | Chikungunya | Zika | Dengue | Chikungunya | Zika |
| | **Total N (%)** | **7,574 (67.5)** | **444 (4.0%)** | **3,205 (28.6%)** | **4,594 (96.5%)** | **91 (1.9%)** | **75 (1.6%)** | **804 (83.7%)** | **140 (14.6%)** | **17 (1.8%)** |
| | **Undifferentiated fever** | 1,232 (18.9%) | 70 (16.9%) | 431 (14.0%) | 2,304 (50.2%) | 57 (62.6%) | 44 (58.7%) | 292 (36.3%) | 28 (20.0%) | 8 (47.1%) |
| | **Dengue** | 4,438 (68.3%) | 48 (11.6%) | 273 (8.9%) | 2,254 (49.1%) | 18 (19.8%) | 18 (24.0%) | 433 (53.9%) | 7 (5.0%) | 1 (5.9%) |
| | **Chikungunya** | 593 (9.1%) | 277 (66.7%) | 384 (12.5%) | 9 (0.2%) | 13 (14.3%) | 1 (1.3%) | 9 (1.1%) | 56 (40.0%) | 0 (0%) |
| | **Non-specified vector-borne diseases** | 239 (3.7%) | 20 (4.8%) | 1,992 (64.7%) | 27 (0.6%) | 3 (3.3%) | 12 (16.0%) | 70 (8.7%) | 49 (35.0%) | 8 (47.1%) |

* SIVIGILA diagnoses: dengue (Codes 210 or 220); chikungunya (Code 217); and Zika (Code 895).

† Healthcare facility's diagnosis is the grouped variable for clinical diagnosis according to the ICD-10 Codes reported on the medical registries

CI = 19.6, 24.3%). Followed by the "Zika/non-specified vector-borne diseases" diagnosis in Cali 14.7% (95%CI = 14.1, 15.3) and Medellin 45.3% (95%CI = 36.6%, 54.1%), and by chikungunya in Villavicencio 10.2% (95%CI = 7.4%, 12.9%). The stratified analysis by healthcare provider in Cali estimated an average probability of reporting of dengue (70.1%; 95%CI = 66.7%, 73.5%), Zika/non-specified vector-borne diseases (35.8%; 95%CI = 28.9%, 42.7%), and chikungunya (6.3%; 95%CI = 4.2%, 8.3%) in the subsidized health care facility. In the contributory health care facility from Cali, the estimated probability of reporting were dengue (20.7%; 95% CI = 20.1%, 21.2%), Zika/non-specified vector-borne diseases (14.1%; 95%CI = 13.5%, 14.6%), and chikungunya (7.8%; 95%CI = 7.3%, 8.2%) (S3 Appendix).

The sensitivity analysis results from the MC-SIMEX showed that the reporting estimates were likely affected by the misclassification of each arboviral disease. Specifically, the observed

**Table 3.  Factors associated to reporting of arboviral diseases in Cali, Medellin, and Villavicencio, 2014–2017.**

| Site | Cali | | Medellin | | Villavicencio | |
|---|---|---|---|---|---|---|
| n | 97,187 | | 32,437 | | 18,121 | |
| Variable | RR | [95% CI] | RR | [95% CI] | RR | [95% CI |
| **Age group** | | | | | | |
| ≤17 Years | Ref. | - | Ref. | - | Ref. | - |
| ≥18 Years | 3.43 | [3.24, 3.64] | 1.51 | [1.43, 1.60] | 1.17 | [1.03, 1.33] |
| **Year** | | | | | | |
| 2014 | Ref. | - | Ref. | - | Ref. | - |
| 2015 | 1.09 | [1.02, 1.18] | 1.11 | [1.02, 1.21] | 0.78 | [0.64, 0.94] |
| 2016 | 2.38 | [2.23, 2.56] | 1.9 | [1.77, 2.04] | 0.81 | [0.66, 0.98] |
| 2017 | 1.71 | [1.57, 1.86] | 0.56 | [0.49, 0.64] | 0.28 | [0.19, 0.41] |
| **Diagnostic** | | | | | | |
| Undifferentiated Fever | Ref. | - | Ref. | - | Ref. | - |
| DENV | 3.6 | [3.44, 3.78] | 5.45 | [5.17, 5.75] | 8.99 | [7.62, 10.61] |
| CHIKV | 1.22 | [1.14, 1.31] | 3.29 | [2.24, 4.63] | 4.17 | [3.08, 5.65] |
| Non-specified vector-borne diseases | 2.42 | [2.28, 2.57] | 5.19 | [3.92, 6.71] | 2.57 | [1.98, 3.32] |

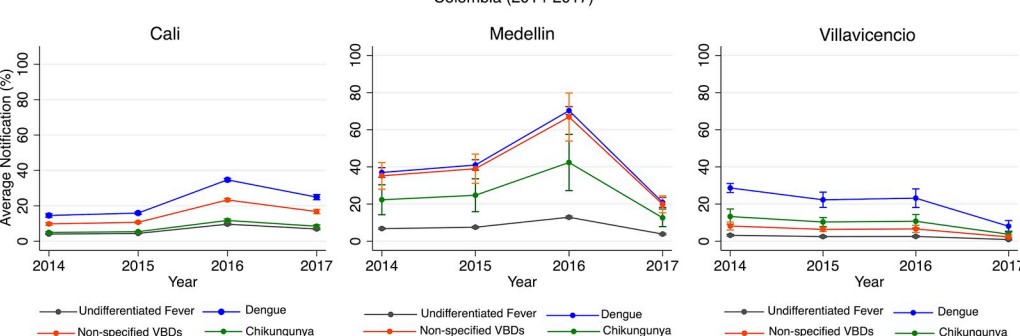

**Fig 2. Adjusted predicted probability and 95% CIs of reporting to SIVIGILA by clinical diagnosis and year of reporting in Cali, Medellin, and Villavicencio (Colombia) 2014–2017.** Non-specified vector-borne diseases (Non-specified VBDs).

(uncorrected) estimates were lower than the extrapolated results (estimates accounting for the possible misclassification), indicating an underestimation of the reporting due to the potential misclassification bias. This was observed consistently in all sites and regardless the parameters used for the simulation extrapolation (Fig 3 and S4 Appendix).

## Discussion

Our capture-recapture study in three different endemicity settings shows that the overall reporting of arboviruses was variable in magnitude and dependent on several factors including age of the case, year of diagnosis, and type of health care provider. Additionally, this study

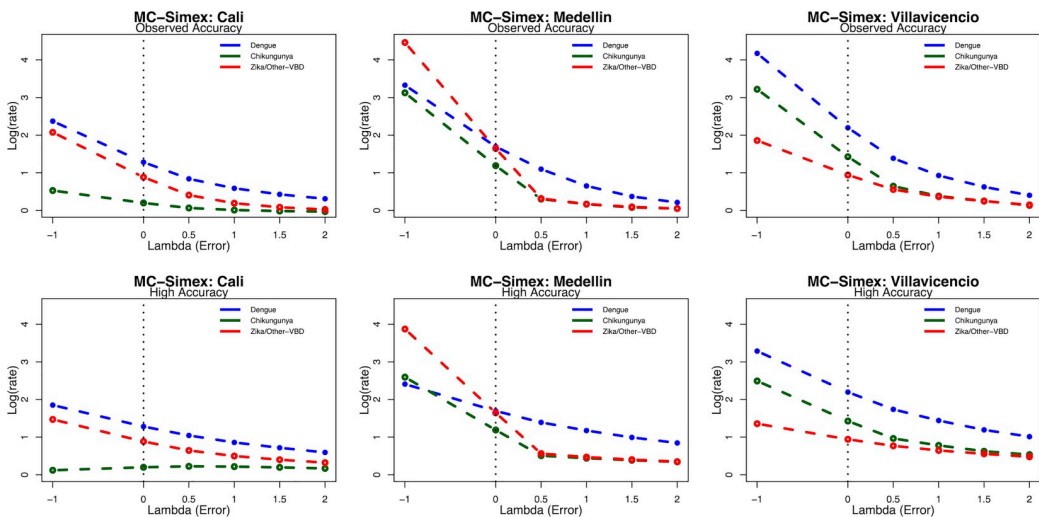

**Fig 3. Simulation Extrapolation results: Measurement Error estimation of the reporting by disease, adjusted by age and year of reporting in Cali, Medellin, and Villavicencio (Colombia) 2014–2017.** The estimated rates on the log scale are presented on the Y axis and the degree/change of misclassification error (Lambda) on the X axis. The dotted line at error level zero in the X axis indicates the observed rates (i.e: uncorrected rates, Table 3) and the estimates at the error level -1, indicates the extrapolated (corrected) rates accounting for the misclassification error. Estimates from error level 0.5 to error level 2, indicate the change in the estimated rates of reporting by adding different levels of misclassification error. Top: results using the observed values of accuracy (sensitivity) for the diagnosis. Bottom: results assuming high accuracy (90% correctly identified) of the clinical diagnosis. Other VBD refers to non-specified vector-borne diseases.

shows the potential effect of misclassification of arboviruses in endemic areas, likely underestimating the burden of these diseases in Colombia.

Although there was not a differential distribution of age across study sites, we observed that cases reported to the national system were older than those who were diagnosed but not reported. Despite the known differences of age distribution among each arbovirus, such as dengue being more observed amongst young patients compared to chikungunya, which is more prevalent in adults and the elderly [1, 11, 12], few other reasons could explain our findings. The limitations associated with the clinical diagnosis, particularly for the pediatric population in absence of confirmatory laboratory tests, could contribute to decrease the likelihood of arboviruses' reporting among individuals under 17 years of age [1, 11, 12, 14, 47, 48, 58]. It is also possible that clinicians initially consider arboviral diagnosis for febrile illnesses in children, although they were later ruled out more systematically and therefore not reported to the surveillance system [12, 14, 25, 48]. Finally, given that dengue is endemic in the three study sites, it is possible that during the study period the population was relatively more susceptible to chikungunya and Zika, diseases that were recently introduced in Colombia and known to be more present among adult populations [24, 34].

Our study showed a differential adjusted rate of reporting by year, which was expected given the endemo-epidemic nature of the arboviruses [1, 2, 29, 37, 47–49, 51]. In 2016, there was a significant burden of dengue and Zika/non-specified vector-borne diseases in Cali and Medellin. These results could be attributed to a true increase in dengue cases in 2016 [15, 35] and also because of the declaration of Zika as public health emergency [24]. Importantly, given the health threat that Zika posed for women of reproductive age and pregnant women and their fetuses, it is possible that an artificial increase in the diagnosis and reporting was observed due to an increase in the health seeking behavior among this population [2, 6]. In Villavicencio, the proportion of reporting between 2014 and 2015 was similar but significantly lower in 2017. The increased rates of reporting rates in 2014 and 2015 for chikungunya coincides with the introduction of the disease in the country and Villavicencio was one of the municipalities most affected by the chikungunya outbreak in the country during that period [15, 29, 34]. Also, it is possible that the symptomatic characteristic of chikungunya [1, 11, 12] could have led to an increased health seeking behavior and therefore an increased reporting during that year [15, 28, 29, 34, 59].

Results from the crude proportion of reporting, the multiplication factors for reporting from the capture-recapture analysis and the estimated adjusted rates of reporting are consistent. These finding indicate higher proportion of reporting in Medellin, followed by Cali and Villavicencio and therefore smaller multiplication factors for reporting in Medellin and Cali, compared to Villavicencio. Likewise, dengue reporting seems higher in all sites, which is consistent to smaller multiplication factors for dengue reporting compared to chikungunya and 'Zika/non-specified vector-borne diseases'. Nonetheless, compared to other arboviral diagnosis, the higher rate of dengue reporting in our study sites was expected. Dengue is a known condition in the country and has been a nationally notifiable disease since 1950 [3, 27, 29, 32]. Compared to chikungunya and Zika, which are newly introduced diseases in our study sites and for which different protocols apply, dengue diagnosis and reporting could present less challenges [1, 24, 34, 47, 51]. Comparatively, the degree of agreement between health care facility's diagnosis and surveillance diagnosis for dengue was better in Cali and Villavicencio. This could be due to the known high burden of dengue in these two cities compared to Medellin, which is considered a city with constant but low presence of dengue during interepidemic periods [27, 35, 42]. Interestingly, a similar pattern of agreement between health care facility's diagnosis and surveillance's diagnosis was observed for chikungunya and 'Zika/non-specified vector-borne diseases' in Cali and Villavicencio as well. Medellin had the highest proportion of "undifferentiated fevers" in the health care facility diagnosis that were later reported to

SIVIGILA as either dengue, chikungunya, or Zika. Similarly, this could be associated with the low number of cases of chikungunya and Zika in Medellin, compared to the presence of these diseases in Cali and Villavicencio and the rest of the country [29, 60].

The differences observed between mainly subsidized or mainly contributory healthcare providers in Cali were mostly of magnitude of effect. Despite that the reporting was higher among people receiving care at the public institution, which provides mostly government subsidized care, the same direction of effects was observed for all predictors (>18 years of age, year 2016, and dengue diagnosis associated to the reporting) across the two types of health care providers.

As anticipated, our simulation extrapolation assessment of misclassification revealed that there was a potential underestimation of arboviruses case reporting. The simulation portion showed that increasing misclassification error moved the estimated reporting rates further towards the null. Consequently, the "error-free" extrapolated estimates were on average, larger than the adjusted but potentially misclassified estimates. Across cities, despite a higher estimated probability rates of reporting, Medellin data seems the most affected by the potential misclassification bias. Across diseases, the potential underestimation of reporting was observed more in chikungunya and for Zika/'non-specified vector-borne diseases' diagnostics than for dengue. These results could be attributed to either the higher burden of arboviruses in Cali and Villavicencio, which could favor a more accurate identification of the arboviruses [27, 29, 59] and or the knowledge of dengue as an endemic condition, compared to the two newly introduced arboviruses [11–14, 36, 58].

These findings pose an important challenge for estimating the arboviral disease burden in Colombia[4]. Importantly, these findings are useful as well to inform forecasting and mathematical modeling exercises in the country and other Latin American endemic settings with similar passive surveillance systems, where the reporting of cases does not rely entirely on laboratory confirmation.

## Limitations

Our study presents novel and robust information about the reporting of arboviruses in three endemic settings of Colombia. However, there are limitations to our study including the lack of information obtained on laboratory confirmation of cases, resulting in limited sensitivity and specificity information. Importantly, some of the healthcare facility data were incomplete and prevented us from a more comprehensive analysis with additional covariates. For instance, detailed information on symptoms, complete blood counts reports, and treatment received could have been used to identify severe arboviral cases and the characterization of reporting by age was not possible due to incomplete individual age data from one healthcare facility. Given the quality of the data, the results were restricted to clinically confirmed diagnosis of arboviruses at healthcare facilities. For Medellin and Villavicencio, it was not possible to conduct stratified analysis by healthcare provider given the absence of information about the type of insurer or because the healthcare facility provided care for different insurance schemes. Finally, for the MC-SIMEX analysis we used three correlation matrixes for the simulation, however, there are several other combinations that could have been used for the correction, each one resulting in different estimates of misclassification [56]. Therefore, simulation extrapolation results should be used as an indicator of the possible effect of misclassification for this specific set of data, given the sensitivity parameter values used in this study [56].

## Conclusion

This complete analysis provides relevant information for the national surveillance system to improve the reporting of arboviruses. This study identifies opportunities to strengthen the

national arbovirus surveillance system by ensuring the reporting of confirmed cases, which depends on the accurate identification (diagnosis) and complete report (documentation) of arboviral diseases at the healthcare facility level (Box 1). This study also supports the need to correct for misclassification when estimating arboviral burden based on clinical diagnosis in endemic settings with passive surveillance systems.

---

### Box 1. Recommendations to strengthen arboviral disease surveillance systems:

- Improve the accurate identification of arboviruses by including lab confirmation in the reporting, when conducted.

- Promote and ensure appropriate reporting forms completion at health care facility level.

- Ensure that surveillance indicators for arboviral reporting are followed adequately.

- Improve the quality of the data by updating surveillance data to capture changes in notifications (change in diagnosis) in a timely manner.

- Periodic verification of healthcare institutions' compliance to following protocols, standards of diagnosis, and reporting of arboviruses.

---

## Supporting information

**S1 Appendix. International Disease Codes and Surveillance codes used in the study.** (PDF)

**S2 Appendix. Case Definitions, according to the National Surveillance Program (SIVI-GILA).** (PDF)

**S3 Appendix. Reporting by health care and predicted probabilities by city.** (PDF)

**S4 Appendix. Simulation Extrapolation for Misclassification results (MC-SIMEX).** (PDF)

## Acknowledgments

We acknowledge the collaboration of the Private and Public Health care institutions providing the recapture data, and the Public Health and the Surveillance offices in Cali, Medellin and Villavicencio, Colombia. We thank Ricardo Gutierrez y Augusto de la Cadena for their collaboration on data collection in Villavicencio.

## Author Contributions

**Conceptualization:** Mabel Carabali, Kate Zinszer.

**Data curation:** Mabel Carabali, Vivian A. Rivera, Neila-Julieth Mina Possu.

**Formal analysis:** Mabel Carabali, Gloria I. Jaramillo-Ramirez, Berta N. Restrepo, Kate Zinszer.

**Funding acquisition:** Kate Zinszer.

**Investigation:** Mabel Carabali, Vivian A. Rivera, Neila-Julieth Mina Possu, Berta N. Restrepo.

**Methodology:** Mabel Carabali, Kate Zinszer.

**Project administration:** Mabel Carabali, Gloria I. Jaramillo-Ramirez, Vivian A. Rivera, Neila-Julieth Mina Possu, Berta N. Restrepo.

**Supervision:** Mabel Carabali, Kate Zinszer.

**Validation:** Mabel Carabali.

**Writing – original draft:** Mabel Carabali.

**Writing – review & editing:** Mabel Carabali, Gloria I. Jaramillo-Ramirez, Vivian A. Rivera, Neila-Julieth Mina Possu, Berta N. Restrepo, Kate Zinszer.

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
