## [Decision Letter · Decision Letter 0]

31 Jul 2020

Dear Dr Carabali,

Thank you very much for submitting your manuscript "Assessing the reporting of Dengue, Chikungunya and Zika to the National Surveillance System in Colombia from 2014-2017: A Capture-recapture analysis accounting for misclassification of arboviral diagnostics." for consideration at PLOS Neglected Tropical Diseases. As with all papers reviewed by the journal, your manuscript was reviewed by members of the editorial board and by several independent reviewers. In light of the reviews (below this email), we would like to invite the resubmission of a significantly-revised version that takes into account the reviewers' comments. 

All three reviewers mentioned that better description of data sources and analysis techniques were needed in the methods section, to better understand the results. Additional concerns about the statistical analyses were raised by multiple reviewers - please consider these comments in your revised version and if you feel that changes to your analyses are not required please justify why. Additionally, please clarify whether data from the same hospital/region can be considered "independent" and if not then if a different regression model (such as poisson regression with generalized estimating equations) would be more appropriate.

We cannot make any decision about publication until we have seen the revised manuscript and your response to the reviewers' comments. Your revised manuscript is also likely to be sent to reviewers for further evaluation.

Sincerely,

Brianna R Beechler, Ph.D., DVM

Guest Editor

Sylvie Alonso

Deputy Editor

All three reviewers mentioned that better description of data sources and analysis techniques were needed in the methods section, to better understand the results. Additional concerns about the statistical analyses were raised by multiple reviewers - please consider these comments in your revised version and if you feel that changes to your analyses are not required please justify why. Additionally, please clarify whether data from the same hospital/region can be considered "independent" and if not then if a different regression model (such as poisson regression with generalized estimating equations) would be more appropriate.

Reviewer's Responses to Questions

**Key Review Criteria Required for Acceptance?**

**Methods**

-Are the objectives of the study clearly articulated with a clear testable hypothesis stated?

-Is the study design appropriate to address the stated objectives?

-Is the population clearly described and appropriate for the hypothesis being tested?

-Is the sample size sufficient to ensure adequate power to address the hypothesis being tested?

-Were correct statistical analysis used to support conclusions?

-Are there concerns about ethical or regulatory requirements being met?

Reviewer #1: Hypotheses are required to structure the paper. This will particularly help with the results and discussion sections. Specifically, it would be interesting to ask the question why the datasets are different. It is not clear from the paper whether the dominant hypothesis is that under-reporting is due to poor reporting to the national surveillance system, clinical diagnosis is poor or whether cross-reactivity/multiple infections are responsible. 

In the methods section, I would like to see the expectations clarified. Do we expect the datasets to be identical if reporting has worked properly? Or is there something more complicated with the laboratory testing process? If a patient goes to a different healthcare facility, will it be reported in the 'recapture' but not the capture?

The choice for statistical testing is unusual, potentially being an inappropriate choice. It is not fully explained, making evaluation difficult. The authors use a Poisson regression on a binary outcome predictor (recaptured or not)? More appropriate to consider a logistic regression with a binomial link function or a Poisson regression on the number of reported cases, with the number available as an offset. 

I also think the presentation of the MC-SIMEX analyses is not ideal, it is presented as a magic black box. What does it do? How does it work? What are the assumptions? Please could you also clearly state the parameters used for the model.

The wording ‘capture-recapture study’ invokes a specific type of analysis, so was surprised to see only a Poisson regression in the methods section.

Reviewer #2: Can you provide any additional details on the healthcare facilities included in the study? Are they public/private? Do they serve particular populations? Do these facilities typically take patients with these diseases or would they have referred the patients to other facilities for confirmation or treatment? If they might refer patients, the facility might expect that other facilities would make the disease report. 

Were individual models fit for each site or all together in one model? It seems they are all together but this could be more clear in the methods section.

Why did you choose to evaluate age as a category instead of as a continuous variable? Why did you decide on these categories?

Reviewer #3: The objectives of the study are clear and the study design is appropriate (using method previously published); the sample size is adequate as well. 

It would be helpful for the methods section to better explain the statistics specifically related to capture-recapture and define the outputs in the Results section more carefully. Even though it is following Vong et al., it would be nice to have some of the basics briefly revisited in the methods section--it's only mentioned in the first sentence of the Methods. For example, are you using Vong's method to estimate the "true" total number of dengue/chik/Zika cases? I don't see that in the Results section at all, but that would be very interesting. While this can be implied from the reporting percents that you show, it could be nice to show the estimated numbers as well. Or maybe I missed that in the results section?

How is rate of notification and IRR that are referred to in Results related to this capture-recapture method? It's not clear how all the statistics that are done fit together into one picture. Also, the sensitivity analysis referred to in the Results section is not well explained in Methods. So, to summarize, the statistical methods need to be significantly expanded and more detail given.

**Results**

-Does the analysis presented match the analysis plan?

-Are the results clearly and completely presented?

-Are the figures (Tables, Images) of sufficient quality for clarity?

Reviewer #1: (No Response)

Reviewer #2: lines 255-257: perhaps this sentence belongs in the methods section.

Figures are very nice.

Reviewer #3: The analysis results don't match what is described in the Methods section, so it is hard to see how everything fits together. The data and conclusions about under-reporting, misclassification, and variation by year and location are very interesting.

Minor comments:

For Table 2 are the Dengue Chikungunya Zika columns total across all years?

Lines 250-257 are not clear--I can't find an explanation of the sensitivity analysis in the Methods. Are you referring to MC-SIMEX?

**Conclusions**

-Are the conclusions supported by the data presented?

-Are the limitations of analysis clearly described?

-Do the authors discuss how these data can be helpful to advance our understanding of the topic under study?

-Is public health relevance addressed?

Reviewer #1: (No Response)

Reviewer #2: For instances when the healthcare diagnosis differed from the SIVIGILA diagnosis, how could this occur? Does the reporting agency make a final reporting diagnosis that might differ from the ICD-10 final clinical diagnosis? Does SIVIGILA review case details for non-specified vector-borne diseases? Does SIVIGILA coordinate with some national testing service data or other test results? Reporting error? Can you discuss how these differences might arise?

Iines 262, 312-316: “…type of healthcare provider” / “difference observed between public and private healthcare providers” I think it is difficult to pinpoint exactly what this effect is, since you have a single healthcare facility for each city (except for Cali) and that isn’t enough healthcare facilities to determine the impact of type of healthcare facility. For your dataset, you can’t really distinguish the impact of city/healthcare facility, as these are essentially the same thing in your data (i.e. one facility per city). It could just be that the particular facility is better/worse at reporting with it not being related to being public or private.

Reviewer #3: The conclusions are supported by the data and analysis and limitations are clear. I think there could be more discussion about how to use this to support forecasting and mathematical modeling by providing reporting rates and thinking about if there are any generalizations that can be made into the future on reporting rates. 

It would also be helpful to better explain the implications of the MC-SIMEX analysis. For example, lines 318-320 are not clear to me--are you saying that you are under-estimating the reporting rates so that reporting is actually better/higher than what you are estimating? Lines 322-323 are also not clear.

In line 281 are you referring to the reporting rate as determined by your capture-recapture stats? It seems like the word reporting is perhaps used differently in different parts of the manuscript...

**Editorial and Data Presentation Modifications?**

Reviewer #1: (No Response)

Reviewer #2: line 25 and throughout: Zika is always capitalized

line 182: change “by” to “at”

Reviewer #3: Zika should be capitalized

line 218 what do you mean by "proportion of reporting" is that proportion of all reported cases?

line 221 what do you mean by "proportion of diagnosis agreement"?

line 229 what do you mean by "notification"? is that different from reporting?

please define IRR

line 236 what do you mean by "adjusted predicted probability of reporting"? adjusted by age only?

**Summary and General Comments**

Reviewer #1: The paper explores an important and well-known problem with dengue, chikungunya and Zika in Latin America. I liked the choice of comparing 3 different cities with varying levels of endemicity and thought that the concept of exploring under-reporting by sex, age, year and healthcare provider was interesting. However, I felt this concept wasn’t fully utilised. If under-reporting in respect of sex, age, year and healthcare provider had been explored for each pathogen then targeted suggestions could have been created. I feel it is really important to separate these factors by pathogen due to their differing host demographics. The paper would benefit from some clear hypotheses to investigate. This would then structure the result and discussion sections better for easier reading. The main concern for the paper is that the methodology is not clearly explained and I am not sure of the choice of statistical testing. This paper is well worth revising for resubmission as it would be interesting to see the concepts of the paper fully explored.

Reviewer #2: Can you clarify in the intro that non-specified vector-borne disease is not reported to SIVIGILA

Reviewer #3: The manuscript could generally benefit by clarification throughout. The data and analysis is very interesting and could be useful both in public health and modeling applications.

PLOS authors have the option to publish the peer review history of their article (what does this mean?). If published, this will include your full peer review and any attached files.

Reviewer #1: No

Reviewer #2: No

Reviewer #3: No
---

## [Decision Letter · Decision Letter 1]

17 Nov 2020

Dear Dr Carabali,

Thank you very much for submitting your manuscript "Assessing the reporting of Dengue, Chikungunya and Zika to the National Surveillance System in Colombia from 2014-2017: A Capture-recapture analysis accounting for misclassification of arboviral diagnostics." for consideration at PLOS Neglected Tropical Diseases. As with all papers reviewed by the journal, your manuscript was reviewed by members of the editorial board and by several independent reviewers. The reviewers appreciated the attention to an important topic. Based on the reviews, we are likely to accept this manuscript for publication, providing that you modify the manuscript according to the review recommendations. 

There are a few remaining minor clarifications needed from one reviewer.

Sincerely,

Brianna R Beechler, Ph.D., DVM

Guest Editor

Sylvie Alonso

Deputy Editor

There are a few remaining minor clarifications needed from one reviewer.

Reviewer's Responses to Questions

**Key Review Criteria Required for Acceptance?**

**Methods**

-Are the objectives of the study clearly articulated with a clear testable hypothesis stated?

-Is the study design appropriate to address the stated objectives?

-Is the population clearly described and appropriate for the hypothesis being tested?

-Is the sample size sufficient to ensure adequate power to address the hypothesis being tested?

-Were correct statistical analysis used to support conclusions?

-Are there concerns about ethical or regulatory requirements being met?

Reviewer #1: The expansion of the methods section has made the paper much clearer. Rather than calling the multiplication factor in lines 221-222 the inverse of the underreporting rate perhaps call it a reporting rate. Also, maybe say here what is a considered a good or bad value for this.

Reviewer #3: methods have been clarified appropriately

**Results**

-Does the analysis presented match the analysis plan?

-Are the results clearly and completely presented?

-Are the figures (Tables, Images) of sufficient quality for clarity?

Reviewer #1: (No Response)

Reviewer #3: also improved and clarified

**Conclusions**

-Are the conclusions supported by the data presented?

-Are the limitations of analysis clearly described?

-Do the authors discuss how these data can be helpful to advance our understanding of the topic under study?

-Is public health relevance addressed?

Reviewer #1: (No Response)

Reviewer #3: appropriate

**Editorial and Data Presentation Modifications?**

Reviewer #1: (No Response)

Reviewer #3: (No Response)

**Summary and General Comments**

Reviewer #1: (No Response)

Reviewer #3: In general, the authors have addressed reviewers' questions and present and interesting analysis of the data.

PLOS authors have the option to publish the peer review history of their article (what does this mean?). If published, this will include your full peer review and any attached files.

Reviewer #1: No

Reviewer #3: No
---

## [Editor Report · Decision Letter 2]

25 Nov 2020

Dear Dr Carabali,

We are pleased to inform you that your manuscript 'Assessing the reporting of Dengue, Chikungunya and Zika to the National Surveillance System in Colombia from 2014-2017: A Capture-recapture analysis accounting for misclassification of arboviral diagnostics.' has been provisionally accepted for publication in PLOS Neglected Tropical Diseases.

Best regards,

Brianna R Beechler, Ph.D., DVM

Guest Editor

Sylvie Alonso

Deputy Editor

---

## [Editor Report · Acceptance letter]

30 Jan 2021

Dear Dr Carabali,

We are delighted to inform you that your manuscript, "Assessing the reporting of Dengue, Chikungunya and Zika to the National Surveillance System in Colombia from 2014-2017: A Capture-recapture analysis accounting for misclassification of arboviral diagnostics.," has been formally accepted for publication in PLOS Neglected Tropical Diseases.

Best regards,

Shaden Kamhawi

co-Editor-in-Chief

Paul Brindley

co-Editor-in-Chief
